# Chitosan (CS)/Hydroxyapatite (HA)/Tricalcium Phosphate (β-TCP)-Based Composites as a Potential Material for Pulp Tissue Regeneration

**DOI:** 10.3390/polym15153213

**Published:** 2023-07-28

**Authors:** Ingrid Zamora, Gilbert Alfonso Morales, Jorge Iván Castro, Lina Marcela Ruiz Rojas, Carlos Humberto Valencia-Llano, Jose Herminsul Mina Hernandez, Mayra Eliana Valencia Zapata, Carlos David Grande-Tovar

**Affiliations:** 1Grupo Biomateriales Dentales, Escuela de Odontología, Universidad del Valle, Calle 4B # 36-00, Cali 76001, Colombia; ingrid.zamora@correounivalle.edu.co (I.Z.); gilbert.morales@correounivalle.edu.co (G.A.M.); carlos.humberto.valencia@correounivalle.edu.co (C.H.V.-L.); 2Laboratorio SIMERQO, Departamento de Química, Universidad del Valle, Calle 13 No. 100-00, Cali 76001, Colombia; jorge.castro@correounivalle.edu.co; 3Grupo de Materiales Compuestos, Escuela de Ingeniería de Materiales, Facultad de Ingeniería, Universidad del Valle, Calle 13 No. 100-00, Cali 760032, Colombia; ruiz.lina@correounivalle.edu.co (L.M.R.R.); jose.mina@correounivalle.edu.co (J.H.M.H.); valencia.mayra@correounivalle.edu.co (M.E.V.Z.); 4Grupo de Investigación de Fotoquímica y Fotobiología, Universidad del Atlántico, Carrera 30 Número 8-49, Puerto Colombia 081008, Colombia

**Keywords:** *Artemia salina*, biocompatibility, biodegradability, endodontic treatment materials

## Abstract

This research focused on developing new materials for endodontic treatments to restore tissues affected by infectious or inflammatory processes. Three materials were studied, namely tricalcium phosphate β-hydroxyapatite (β-TCP), commercial and natural hydroxyapatite (HA), and chitosan (CS), in different proportions. The chemical characterization using infrared spectroscopy (FTIR) and X-ray diffraction (XRD) analysis confirmed the composition of the composite. Scanning electron microscopy (SEM) demonstrated that the design and origin of the HA, whether natural or commercial, did not affect the morphology of the composites. In vitro studies using *Artemia salina* (*A. salina*) indicated that all three experimental materials were biocompatible after 24 h, with no significant differences in mortality rate observed among the groups. The subdermal implantation of the materials in block form exhibited biocompatibility and biodegradability after 30 and 60 days, with the larger particles undergoing fragmentation and connective tissue formation consisting of collagen type III fibers, blood vessels, and inflammatory cells. The implanted material continued to undergo resorption during this process. The results obtained in this research contribute to developing endodontic technologies for tissue recovery and regeneration.

## 1. Introduction

The different tissue defects caused by diseases and trauma have motivated various groups to research new composite sources that allow the regeneration of injured tissues and organs [1]. As a result, tissue engineering has led to the formation of new scaffolds with a high impact in regenerative dentistry, aiming to simulate the desired properties of the affected site [2].

The pulp tissue of teeth can be affected by bacterial invasion or dental trauma. Usually, the tissue becomes inflamed, and if no action is taken to interrupt the inflammation, pulp necrosis occurs, promoting the death of odontoblasts and interrupting root development in immature teeth [3]. In recent years, regenerative endodontics has presented new possibilities for treating necrotic immature permanent teeth by developing new pulp tissue through the combination of three essential elements for tissue regeneration: stem cells, growth factors, and scaffolds [4]. In this regard, different proposals have been used based on materials that lead to the stimulation of cells and tissues by incorporating chemical substances such as calcium hydroxide and mineral trioxide aggregates (MTAs) [5]. However, in many cases, during regenerative endodontic procedures, calcium hydroxide and MTA-based materials tend to induce the formation of mineralized nodules [6].

MTAs consist of calcium, aluminum, and selenium and have become the most studied endodontic material in the last two decades due to their biocompatibility, hydrophilicity, bioactivity, radio-opacity, sealing capacity, and low solubility [7,8]. However, the main disadvantages of using MTAs are the long setting time, microhardness, handling difficulty, high cost, and discoloration of the remaining tooth structure [9]. Therefore, new substitutes are being sought to solve this problem, such as mineralized agents like bioceramics based on calcium phosphate or highly bioactive polymers like chitosan.

The best-known calcium-phosphate-based bioceramics are hydroxyapatite (HA) (Ca_10_(PO_4_)_6_(OH)_2_) and tricalcium phosphate (β-TCP) (Ca_3_(PO_4_)_2_). A solid-state reaction involving thermal decomposition is generally used to prepare β-TCP powders. At the same time, HA is synthesized through four different methods: (i) dry methods; (ii) wet methods; (iii) the microwave (MW)-assisted method, ball-milling, or ultrasound; and (iv) various other methods [10]. It has been found that β-TCP is quite soluble in body fluids, while HA is relatively stable [11]. Additionally, β-TCP is considered an osteoconductive material with a reasonable resorption rate, making it ideal as a bone substitute [12]. Likewise, synthetic HA is considered suitable but has a slower resorption rate [13]. In this sense, different investigations have focused on studying the resorbability conditions of the biphasic bond between HA and β-TCP according to the proportion of each component, the pH of the medium, and the crystallinity for the design of scaffolds [14,15]. Therefore, it is necessary to systematically produce two-phase ceramics with different HA/β-TCP ratios.

On the other hand, chitosan (CS) is a natural polysaccharide derived from the deacetylation of chitin, the main component of the exoskeleton of crustaceans, arthropods, and the mycelium of many fungi [16]. This natural polymer features intrinsic properties such as high hydrophilicity, antibacterial activity, biocompatibility, and high biodegradability and is suitable for producing natural ingredients and drug delivery systems such as hydrogels, membranes, nanofibers, and microspheres [17,18,19]. However, like many polysaccharides, chitosan has the disadvantages of low solubility, poor stability in physiological fluids, and poor mechanical properties [20]. To compensate for these disadvantages, CS has been blended with different synthetic polymers or mineralized salts, such as the ternary mixture CS/HA/β-TCP, which showed that the crosslinking process influenced the physicochemical and mechanical properties. In addition, this composite demonstrated a more significant interaction between HA and β-TCP due to the presence of CS, which facilitated a uniform distribution in the scaffold [21].

Additionally, this ternary blend has shown good swelling, mechanical, and thermal properties related to the HA/β-TCP ratio. Cell culture experiments demonstrated that the number of L929 and Saos-2 cells on the surface of the composite scaffold increased, and that a higher HA/β-TCP ratio stimulated cell growth [22]. To date, there have been no studies on subdermal implantation considering the biocompatibility of the material due to the ternary mixture CS/HA/β-TCP, taking into account the possible variations in the physicochemical properties due to the implementation of natural or synthetic HA. The results of in vitro and in vivo biocompatibility evaluations have demonstrated the potential of these materials in the field of tissue engineering due to the emergence of connective tissue and the decrease in inflammatory response.

## 2. Materials and Methods

### 2.1. Materials

All reagents used for this research were commercial-grade. For the formation of the solid composite, we used HA powder (tri-basic calcium phosphate) (HCa_5_O_13_P_3_) with a molar mass of 502.31 g/mol and β-TCP (Ca_3_O_8_P_2_) with a purity grade of ≥96.0% and molar mass of 310.18 g/mol, which were obtained from Sigma Aldrich (Palo Alto, CA, USA). In addition, calcium chloride (CaCl_2_) with a purity grade of ≥96.0% was obtained from the same company to prepare the liquid phase. A capillary viscometer was used to determine the molecular weight of the CS, and gel permeation chromatography (GPC) was applied with viscous solutions of CS, allowing the measurement of the viscosity average molecular weight (Mv). For this purpose, the drop times of 25 mL solutions prepared from a standard solution (Fungilab, Santiago de Chile, Chile) were measured with an Ubbelohde viscometer.

The intrinsic viscosity is the intercept value of the specific viscosity curve and is symbolized by [*η*]. The Houwink–Sakurada equation (Equation (1)) [23] relates this viscosity to the constants *K* (0.074 mL/g) and a (0.76) estimated at 25 °C in a mixture of 0.3 M acetic acid and 0.2 M sodium acetate [24], and the Mv calculated for CS was 144,000 Da.
(1)$$\left[\eta \right]=K{\left(Mv\right)}^{a}$$

The degree of CS deacetylation was 89–90% and was determined by 400 MHz ^1^H-NMR spectroscopy using a BRUKER AVANCE II (Bruker, Berlin, Germany) with 300 K temperature control and a Thermo Electron Flash EA 1112 elemental analyzer (Thermo Fischer, Waltham, MA, USA). Measurements were performed using D_2_O, two drops of trifluoroacetic acid as the solvent, and 3-(trimethylsilyl) propionic acid-d4 as the internal standard.

### 2.2. Synthesis of Natural Hydroxyapatite

The natural hydroxyapatite of bovine origin was obtained according to previous research [25]. The extraction of apatite began with the collection of bovine femur and hip bones. These were first cut into pieces approximately 4 to 5 cm wide. The bones were then brought to 103 °C to facilitate the separation of the bone tissue from the attached medullary, fat, and muscle tissues. This cooking process was repeated until the cooking water showed no traces of fat. Afterward, the bones were oven-dried for 24 h at 40 °C. Finally, the particle size reduction process was initiated with a jaw mill (Idar-Oberstein, Germany). This was followed by a knife mill (Idar-Oberstein, Germany), starting with a giant screen (4000 microns) and proceeding to the smallest screen (250 microns), and the process ended with a ball mill (Idar-Oberstein, Germany). Next came the calcination process, in which the apatite was fed into the furnace (Nabertherm LHT 02/18, Lilienthal, Bremen, Germany) at a ramp rate of 5 °C/min up to 900 °C, at which temperature it was held for 2 h.

### 2.3. Preparation of the CS/HA/β-TCP Blocks

To obtain a composite material with good structural integrity and sound osteoconductive reinforcement, small amounts of HA were used so as not to affect the composite’s resorption processes, alongside small amounts of CS so as not to induce strong inflammatory responses. We prepared the CS/HA/β-TCP blocks in two stages. The first stage involved physically mixing the different solid components according to their percentage by weight in the scaffold. Once the mixture was obtained, a suspension of 2.5 g was acquired from the solid composite mixture for each formulation, and a 5.4 wt% solution of CaCl_2_ was added, as shown in Table 1.

### 2.4. Characterization of the CS/HA/β-TCP Blocks

#### 2.4.1. Fourier Transform Infrared Spectroscopy FT-IR

The determination of the individual functional groups present in the CS/HA/β-TCP blocks was performed by FT-IR on an Affinity-1 IR infrared spectrophotometer (Shimadzu, Kyoto, Japan) in the range 500–4000 cm^−1^ with a 4 cm^−1^ spectral resolution in transmittance mode using the attenuated total reflectance (ATR) method with a diamond tip.

#### 2.4.2. X-ray Diffraction (DRX)

The presence of the different crystalline planes in the CS/HA/β-TCP blocks was determined employing a PANalytical X’Pert PRO MRD diffractometer (Malvern Panalytical, Jarman Way, Royston, UK) using copper radiation at a wavelength of Kα1 (1.540598 Å) and Kα2 (1.544426 Å). The diffractometer was operated in the secondary electron mode at 45 kV in the 2θ range of 5 to 80° at a scan rate of 2 degrees/min with a scan speed of 2.63 s and a step size of 0.01°.

#### 2.4.3. Scanning Electron Spectroscopy (SEM) and Energy Dispersive Spectroscopy (EDS) for the CS/HA/β-TCP Blocks

The morphology of the CS/HA/β-TCP blocks was studied by scanning electron microscopy using a JEOL JSM-6490LA microscope (Musashino, Tokyo, Japan) operated in the secondary electron mode with an accelerating voltage of 20 kV. The samples were doped with gold to increase their conductivity. In addition, the chemical microanalysis of small areas of the CS/HA/β-TCP blocks was carried out using an Oxford Instrument Model INCAPentaFETx3 probe (Abingdon, UK) for energy dispersive spectroscopy (EDS) at a resolution of 15 nm, an accelerating voltage of 1 kV, and a working distance of 6 mm in the secondary electron mode.

#### 2.4.4. In Vitro Viability of the CS/HA/β-TCP Blocks

The *A. salina* test allows a preliminary approximation of a material’s toxicity and is standardized and recognized in the literature [26]. Preliminary cytotoxicity studies were performed using the *A. salina* (AS) test (Brine shrimp direct, Ogden, UT, USA), and the AS was hydrated with two liters of 38.5 wt% NaCl saline solution with aeration and permanent light. After 48 h of incubation, nauplii were transferred to 24-well culture boxes (Corning, Thermo Fischer Waltham, MS, USA), with 5 mL of seawater solution per well and quintuplicate samples of the three formulations of the material [27].

Ten larvae of *A. salina* with an age of 48 h were added to each well. Bonzyme enzyme soap (Eufar Laboratories, Bogota, Colombia) was used as a positive control, and artificial seawater solution was used as a negative control. After 24 h of incubation, the results were collected by performing the same procedure five times for each formulation. The mortality rate was determined according to the number of dead *A. salina* larvae and the number of initial larvae through the following equation [28]:(2)$$\mathrm{Mortality}\left(\%\right)=\frac{\mathrm{Number}\;\mathrm{of}\;\mathrm{AS}\;\mathrm{death}}{\mathrm{Number}\;\mathrm{of}\;\mathrm{As}\;\mathrm{Initial}}\times 100$$

#### 2.4.5. In Vivo Biocompatibility Study of CS/HA/β-TCP Blocks

For the in vivo biocompatibility test, six male Wistar rats (*Rattus norvegicus* domestic) that were five months old and had an average weight of 380 g were randomly distributed into two groups for observations at 30 and 60 days, taking into account that the minimum number of rats for reliable results is three according to the ISO 10993-6 standard [29]. This made it possible to apply Rusell and Burch’s principle of the three Rs [30].

Once the implantation periods of 30 and 60 days were completed, the biomodels were euthanized by the intraperitoneal injection of sodium pentobarbital/sodium diphenylhydantoin (0.3 mL/biomodel kg) (Euthanex, INVET Laboratory, Cota, Colombia); the samples were fixed in buffered formalin for 48 h and processed following standardized routine procedures to obtain histological sections of 6 μm [31,32]. These sections were subjected to histological staining techniques with hematoxylin-eosin (HE), Masson’s trichrome (MT), and Gomori’s trichrome (GT). Histological images were taken with a Leica DM750 optical microscope and a Leica DFC 295 camera. The images were processed with Leica Application Suite version 4.12.0 software (Leica Microsystem, Mannheim-Germany). The procedures were performed using the recommendations of the ARRIVE (Animal Research: Reporting of In Vivo Experiments) guide; no biomodel deaths or post-surgical complications occurred in the research. The ethical review was overseen by the Ethics Committee for Biomedical Animal Experimentation (CEAS) (Cali, Colombia) according to Resolution No. CEAS 006-022.

#### 2.4.6. Statistical Analysis

The biocompatibility results for the interaction with *A. salina* are presented as the mean value of at least five replicates ± SD. Analysis of variance (ANOVA) and Tukey’s method at a 95% confidence level (α = 0.05) were used to assess the significance of the effect of changing the formulation on the in vitro characterization responses of the blocks. Statgraphics Centurion XVI was used for statistical analyses.

## 3. Results and Discussion

### 3.1. FT-IR Spectroscopy

An FT-IR analysis was performed to determine the functional groups of each component in the composite, as shown in Figure 1. We observed no significant variations in the chemical composition of the different formulations, since both formulations with commercial hydroxyapatite (Formulations 1 and 2) and the formulation with the experimental natural apatite (Formulation 3) presented similar spectra. In this context, the stress vibrational modes of the ${PO}_{4}^{-3}$ bond were located at 936 and 1100 cm^−1^; the symmetrical stress band at 3400 cm^−1^ was related to the OH group originating from the water absorbed by the environment; and, finally, the band located at 1020 cm^−1^ overlapped with the triple-degenerate asymmetric stretching of the PO_4_ group in hydroxyapatite [33] and the vibrational mode of the C-O-C bond in chitosan [34]. The formulations showed a band overlap due to the presence of similar groups, such as OH and PO_4_, in HA and β-TCP.

On the other hand, the weak band located at 936 cm^−1^ was mainly assigned to the P-O-H vibrations found in the HPO_4_ group of the HA. However, this vibrational mode could also have been due to the contribution of the C-O and CO_3_ groups in both the β-TCP and HA [11]. Different authors have reported that the weight contribution of each material in β-TCP/HA mixtures is affected by the natural or commercial origin of the HA, as well as to the ratio in weight between the β-TCP and HA, as reflected in the following formula: Ca_10-x_[(HPO_4_)_x_(PO_4_)_6-x-y_(CO_3_)_y_][(OH)_2-x-y_(CO_3_)_y_] [35]. The FTIR spectra of all three formulations indicated that the characteristic bands of CaP and CS were present in the composites. Additionally, these results suggested that the origin of the HA did not matter, because no significant differences were found in the FT-IR spectra of the three formulations. Therefore, with the weight percentages in the composites kept constant, no changes in the intensities of the PO_4_ or CS groups were observed, as in previous research [21,22].

### 3.2. X-ray Diffraction

Figure 2 shows the XRD pattern of the CS/HA/β-TCP blocks. In the diffractogram, the characteristic diffraction peaks of the β-TCP structure were observed, which presented 2θ angles of 13.6, 17.0, 21.8, 25.7, 26.5, 27.8, 29.6, 32.4, 35.6, 39.8, 41.1, and 41.7° in relation to reflection of the crystallographic planes (004), (110), (113), (106), (200), (202), (008), (205), (215), (217), (224), and (303), respectively, the latter being the most representative to confirm a tetragonal phase with a 76-P41 space group according the international standard JCPDF 00-020-0024 [36]. Additionally, it was observed that some HA angles overlapped with those belonging to the β-TCP. In this regard, we determined 2θ angles of 10.8, 21.8, 31.0, 34.4, 35.6, 47.0, 48.0, 48.4, 51.5, and 53.0° attributed to the crystallographic planes (010), (020), (121), (022), (031), (222), (132), (230), (140), and (004), respectively, which were consistent with the reference code ICCD 98-010-5020 for a hexagonal structure with space group P 63/m.

On the other hand, previous studies have reported six polymorphisms for chitosan: tendon, annealed, 1-2, L-2, form I, and form II. Furthermore, for all formulations, 2θ angles of 10 and 20° were observed, attributed to the (122) and (110) planes with a semi-crystalline nature [37]. However, according to these results, the crystallinity of CS is affected by the presence of HA/β-TCP, because this bicomponent mixture weakens the intramolecular interactions of the chitosan chain [38]. This observation is similar to those of other previously reported works [39,40,41].

### 3.3. SEM-EDS

Micrographic studies of the CS/HA/β-TCP composites are shown in Figure 3. The surface micrographs presented an agglomerated and spherical structure. In general, the grain size for such composites depends on the calcination temperature at which they are prepared [42,43]. However, in our case, as the material was not calcined, no trend in grain size was found. Therefore, it could be said that the composition and origin of the HA, whether natural or commercial, did not affect the morphology of the composites.

The percentage content of each element was determined by energy-dispersive coupled X-ray spectroscopy (EDS), as shown in Table 2 and Appendix A. In general, the analyses indicated that the composites mainly consisted of C, O, P, and Ca for all formulations, with a small amount of Cl (1.54%) for the F2 formulation, probably influenced by the higher percentage of the liquid phase comprising a calcium chloride solution. On the other hand, the F1 and F3 formulations did not show significant differences in terms of the proportion of elements present, suggesting that commercial apatite and natural bovine apatite performed similarly in this respect.

### 3.4. Results of Biocompatibility Test with A. salina

Figure 4 presents the results of the toxicity bioassays using *A. salina*. This bioassay could be considered a preliminary evaluation of the toxicity using *A. salina*. In this regard, samples with an LC_50_ above 1000 µg·mL^−1^ are considered non-toxic, while samples with an LC_50_ below 100 µg·mL^−1^ are considered highly toxic [26].

Dilutions containing *A. salina* were prepared and transferred to the wells containing the different formulations (Figure 4). Generally, the dispersion of the particulate material was observed, particularly in formulations F1 and F2. Nauplii that were observed to be completely immobile were counted as dead, and mobile nauplii were considered alive.

Figure 4 presents the observations after 24 h of incubation with *A. salina*. Many particles of the materials and several nauplii swimming in contact with them were observed, and a few dead nauplii were seen in all wells. In the positive control wells, all nauplii were found dead.

The observation of the wells at 24 h showed that most of the larvae were found to be alive, with the highest percentage of mortality for the F1 formulation at 22%, slightly above that of the negative control group (18%) but still below the 30% value (above which a material is considered toxic in this test) [44]. The results of the analysis showed that all three experimental materials presented biocompatibility at 24 h, with no significant differences found between the groups in terms of the mortality rate (*p* < 0.0001) (Figure 5).

### 3.5. In Vivo Biocompatibility Test for CS/HA/β-TCP Composites

Before retrieving the samples of the implanted tissues, the macroscopic observation of the implantation area was carried out following the recommendations of the UNE-EN ISO 10993-6 standard [29].

All biomodels presented a similar appearance after 30 and 60 days of implantation, with complete hair regrowth and the absence of fistulas and purulent tissue (Figure 6).

#### 3.5.1. Histological Results for the F1 Formulation at 30 and 60 Days

The results of the histological analysis 30 and 60 days after implantation are shown in Figure 7. At 30 days after implantation, the Masson’s stain showed the presence of a fibrous capsule composed of collagen type I fibers at the implantation site (Figure 7A). The material was fractionated inside this capsule and surrounded by collagen type I fibers (Figure 7B) with inflammatory cells and numerous blood vessels (Figure 7C).

At 60 days after implantation, the fibrous capsule surrounding the implanted material was still present, but a more significant amount of collagen fibers was observed (Figure 7D), as well as the presence of a collagen septum, which indicated the progression of the healing process towards the interior of the implanted area (Figure 7E). In addition, at 100× magnification, numerous blood vessels were observed in addition to the fibers (Figure 7F).

Generally, the presence of a fibrous capsule during the scarring process indicates a foreign body reaction process, whereby the body attempts to limit the damage caused and allow tissue restoration. However, after the implantation of the materials, a series of events led to a cicatricial foreign body reaction, in which the implanted material was encapsulated by inflammatory cells [45], as shown in Figure 7.

The degradability of an implanted material is influenced by the percentage ratio of each component [12]. The F1 implant material consisted mainly of 87.7% β-TCP and 8.8% HA, considered osteoconductive, degradable, and resorbable. In addition, F1 also contained 3.5% CS, which has demonstrated excellent biocompatibility and biodegradability due to the content of β-1,4-N-acetylglucosamine [16]. Since biodegradability is a requirement of interest in biomedical applications, this work proposed the combination of β-TCP, HA, and CS to regulate biodegradability while maintaining biocompatibility. Furthermore, the histological results showed a material in the process of resorption, with the presence of a fibrous capsule and the persistence of material remnants at 60 days.

#### 3.5.2. Histological Results for the F2 Formulation at 30 and 60 Days

The observations for the F2 formulation at 30 and 60 days of implantation were very similar to those reported for F1, with the difference being the appearance of type III collagen fibers. At 30 days of implantation, remnants of the implantation surrounded by a fibrous capsule (Figure 8A) were observed amidst an inflammatory infiltrate (Figure 8B). In addition, Gomori trichrome staining showed the presence of type III collagen fibers in the capsule and between the particles (Figure 8C).

At 60 days post-implantation, the fibrous capsule decreased in size (Figure 9D), and Masson’s trichrome staining showed the presence of type I collagen fibers among the remaining implanted material, with the presence of inflammatory cells (Figure 8E,F).

The F1 and F2 formulations contained CS/HA/β-TCP components in the same proportions, and both included CaCl_2_ as a liquid phase; however, in F1, the proportion of this component was 0.8%, while in F2, the proportion of CaCl_2_ increased to 1%. The increase in the CaCl_2_ percentage influenced the healing response: the histological images for the F2 formulation seemed to indicate a more advanced healing process when compared to the results of the F1 formulation, as manifested by a decrease in capsule thickness and the more rapid resorption of the implanted material (Figure 9). Collagen types I and III are common in skin healing; both types are expected to be present initially. However, collagen type III is mostly degraded and replaced by collagen type I in the final healing phase [46].

#### 3.5.3. Histological Results for Formulation F3 at 30 and 60 Days

At 30 days after implantation, Masson’s trichrome staining showed that the material was surrounded by a capsule of type I collagen fibers (Figure 9A), implanted material was present amidst an inflammatory infiltrate (Figure 9B), and particles of the material were surrounded by bundles of collagen fibers and covered by connective tissue with the presence of type I collagen fibers (Figure 9C).

At 60 days, a fibrous capsule structure with remnant material was observed (Figure 9D). At 10× magnification, the fibrous capsule was revealed to comprise tissue made up of type I collagen fibers that occupied the area where the material was implanted; parts of the material were also observed to be in the process of degradation/reabsorption (Figure 9E). Meanwhile, at 100× magnification, the area with remnant material was occupied by some particles of the material, which were surrounded by tiny collagen fibers and inflammatory cells covered by connective tissue (Figure 9F). The later result was consistent with the stage of tissue remodeling in which the previously deposited clumps of type III and II collagen fibers are progressively degraded and replaced by type I collagen fibers [47].

The inflammatory response observed in the scarring process of the F3 formulation was slightly different from that of the other two formulations. For F3, the capsule appeared to be in the process of disintegration and replacement by connective tissue; thus, at 30 days, a structure compatible with a thin fibrous capsule was observed (Figure 9A) that appeared to increase in size by the 60-day timepoint. However, the images at 10× and 100× indicated a decrease in the remaining material and its coverage by connective tissue. This process seemed to be influenced by the presence of the natural apatite.

Any biomaterial implanted in living tissue incites an inflammatory reaction, including a foreign body reaction and the fibrous encapsulation of the material [48]. In the case of resorbable materials, the capsule will persist as long as the material exists; furthermore, as the implanted material is degraded/reabsorbed by the inflammatory cells, the foreign body reaction will resolve [49]. In general, the behavior was very similar for the three formulations, as a foreign body reaction with capsule formation around the implanted material was observed; furthermore, we observed that collagen fibers enwrapped groups of particles. Gomori and Masson’s trichrome staining identified that the capsules were composed of collagen I and III fibers (Figure 7, Figure 8 and Figure 9), which agreed with what has been reported in the literature [45,46].

Although the three formulations performed similarly regarding the overall healing mechanism, it was possible to observe differences between them. For example, formulation F1 at 60 days presented many remaining particles amidst many inflammatory cells (Figure 7). Still, the presence of particles and cells decreased in formulation F2 for the same period (Figure 8). 

Another important observation was the size of the area occupied by the collagen fibers (type I and type III). While in the F1 formulation, a fibrous capsule and clusters of fibers surrounding the particulate material were observed, in formulation F2, the capsule was smaller. In formulation F3, at 30 days, the capsule was still observable (Figure 9A), but at 60 days, the capsule was replaced by connective tissue with a high content of type I collagen extending over the implantation zone (Figure 9E).

In the scarring process, the temporary matrix formed from plasma proteins is degraded and replaced by bundles of type I and III collagen fibers, which surround the implanted material, as observed in this investigation (Figure 8A,B). Subsequently, the remodeling of the extracellular matrix occurs. Finally, these collagen fiber bundles are replaced by type I collagen, consistent with what was observed in Figure 9, where simultaneously with the degradation/resorption of the material, the capsule disappeared, and connective tissue with abundant type I collagen fibers was formed (Figure 9D). Interestingly, many blood vessels and collagen fibers developed during the healing process, especially in the F1 formulation.

At the beginning of the scar response, the provisional matrix is formed from blood proteins; in addition, a process of hypoxia occurs, which, in conjunction with the healing cascade and the release in histamine and growth factors, such as the vascular endothelial growth factor (VEGF), stimulates angiogenesis. This process usually occurs in the first few days of healing and subsequently as the provisional matrix is replaced by the extracellular matrix [50]; however, for the F1 formulation, the abundant presence of blood vessels was observed even within 60 days, in contrast to what was observed for the other formulations.

Given the above, these approximations regarding the behavior of the different formulations provided an idea as to the processes that may occur in the human body once the blocks are implanted in subdermal tissue, which resembles pulp tissue, for regenerative evolution. However, the in vitro study employing *A. salina* and the subdermal biocompatibility assay using biomodels require complementary cytotoxicity studies with pulp cells and in vivo tests using a dental animal model to approach the pulp object of this study.

## 4. Conclusions

This work successfully reported the synthesis of three CS/HA/β-TCP composites, and their biocompatibility was evaluated in vivo and in vitro using biomodels and *A. salina*, respectively. Additionally, the presence of each HA and β-TCP component was observed through the characteristic PO_4_ bands of the mixture of the inorganic compounds that were blown together. On the other hand, the surface architecture showed that no significant differences were found, because the morphology of the formulations did not change, suggesting a lack of dependency on the origin of the HA, either commercial or natural. Additionally, according to the SEM-EDS results, in contrast to the F2 formulation, it was observed that there was no significant difference in terms of the proportions of the components between F1 and F3. The CS/HA/β-TCP composites were shown to be biocompatible in the subdermal implantation model at 30 and 60 days in the biomodels, allowing healing processes to occur via tissue architecture recovery without necrotic processes. In the resolution process, the material was fragmented and reabsorbed via phagocytic mechanisms, causing the appearance of inflammatory cells. This promoted the formation of collagen type I and III, as seen using the techniques of Gomori and Masson; blood vessels; and some inflammatory cells, which continued with the reabsorption process of the material.

## Figures and Tables

**Figure 1 polymers-15-03213-f001:**
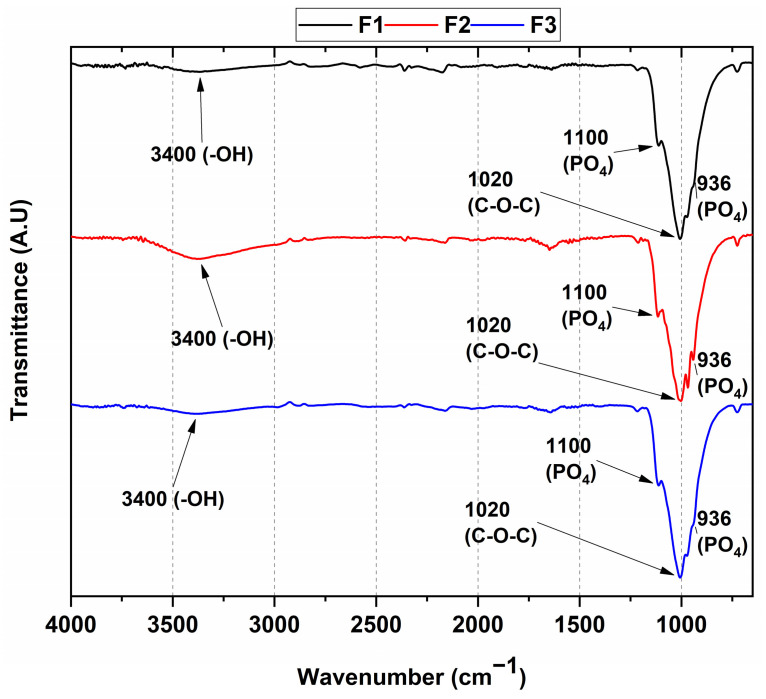
ATR-FTIR of the CS/HA/β-TCP composites. F1—87.7% β-TCP/3.5% CS/8.8% commercial HA/0.8 mL CaCl_2_; F2—87.7% β-TCP/3.5% CS/8.8% commercial HA/1 mL CaCl_2_; F3—87.7% β-TCP/3.5% CS/8.8% natural HA/0.8 mL CaCl_2_.

**Figure 2 polymers-15-03213-f002:**
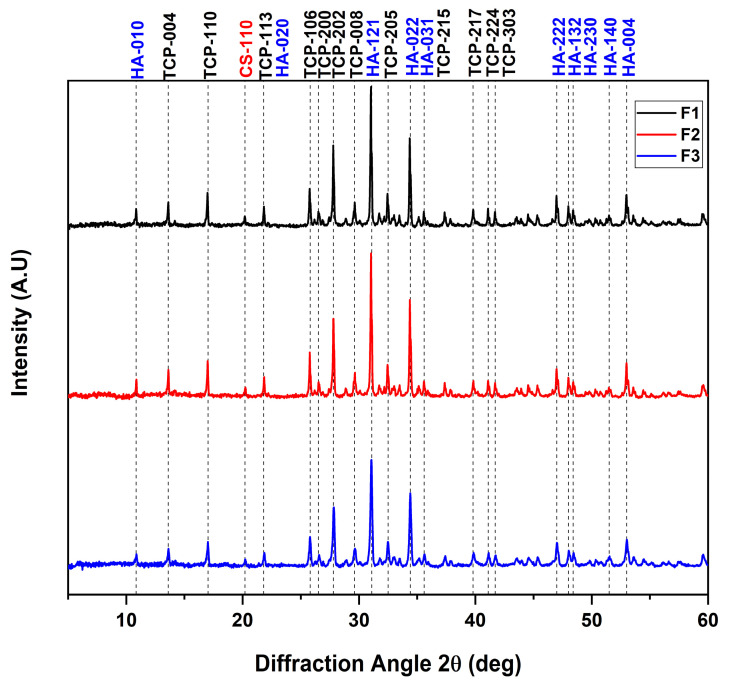
Diffractogram of the CS/HA/β-TCP composites. F1—87.7% β-TCP/3.5% CS/8.8% commercial HA/0.8 mL CaCl_2_; F2—87.7% β-TCP/3.5% CS/8.8% commercial HA/1 mL CaCl_2_; F3—87.7% β-TCP/3.5% CS/8.8% natural HA/0.8 mL CaCl_2_.

**Figure 3 polymers-15-03213-f003:**
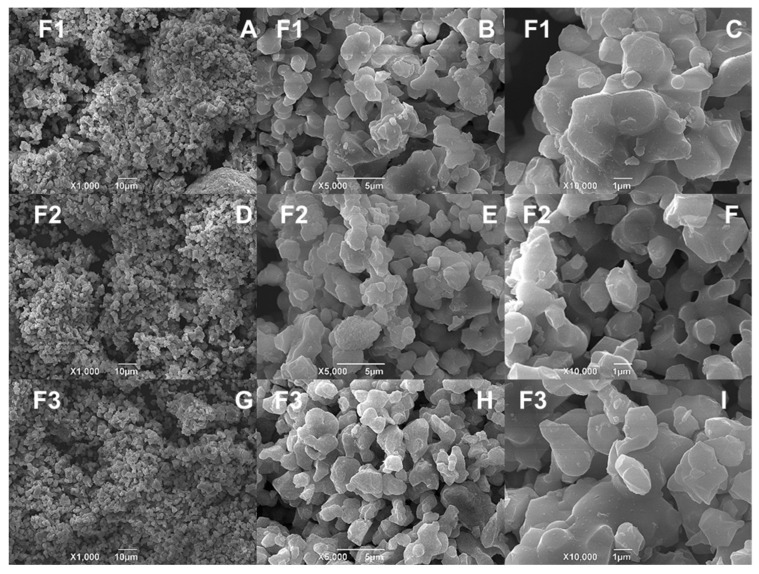
Morphology of CS/HA/β-TCP composites according to SEM. F1 (**A**) at 1000× and 10 µm, (**B**) at 5000× and 5 µm, and (**C**) at 10,000× and 1 µm; F2 (**D**) at 1000× and 10 µm, (**E**) at 5000× and 5 µm, and (**F**) at 10,000× and 1 µm; F3 (**G**) at 1000× and 10 µm, (**H**) at 5000× and 5 µm, and (**I**) at 10,000× and 1 µm.

**Figure 4 polymers-15-03213-f004:**
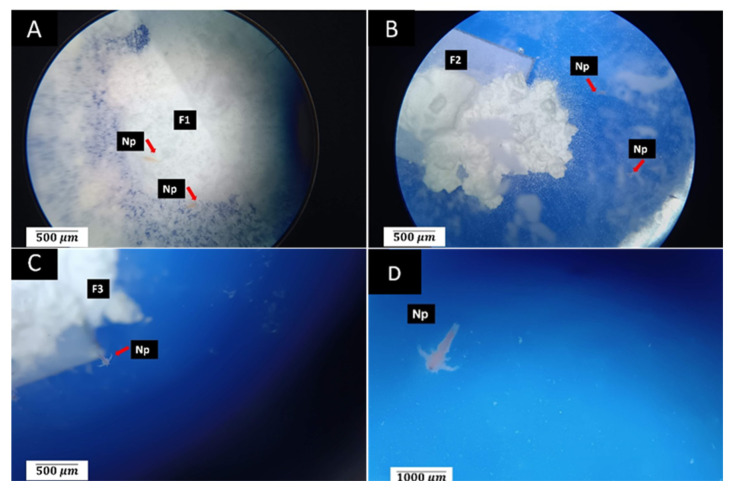
Observation of nauplii after 24 h of exposure to the formulations: (**A**) sample F1 at 4×; (**B**) sample F2 at 4×; (**C**) sample F3 at 3×. Stereo-optical microscopy technique: (**D**) control at 4×. F1—formulation sample F1; F2—formulation sample F2; F3—formulation sample F3; Np—nauplius.

**Figure 5 polymers-15-03213-f005:**
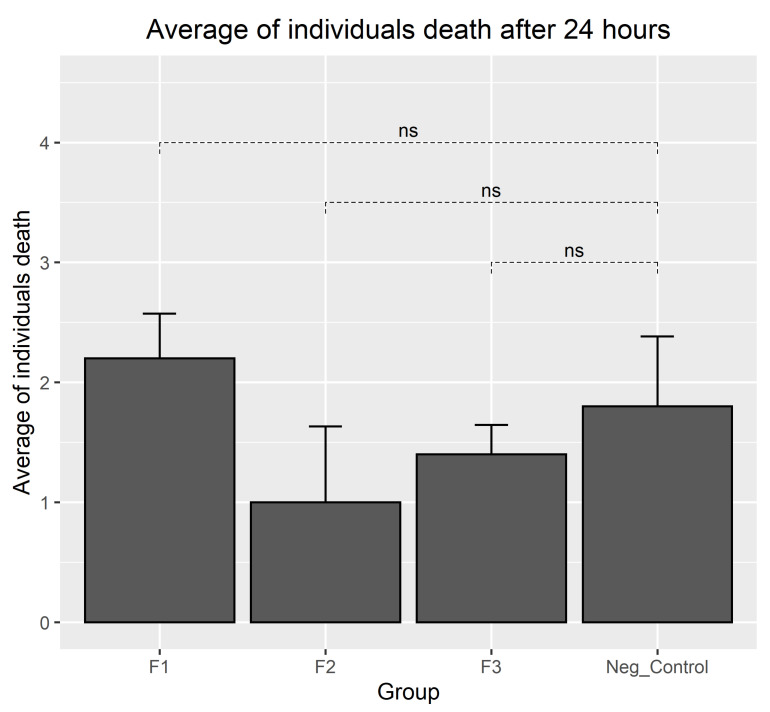
Number of dead A. salina individuals after 24 h (ns: non-significant differences between treatments and trials; *p* < 0.01).

**Figure 6 polymers-15-03213-f006:**
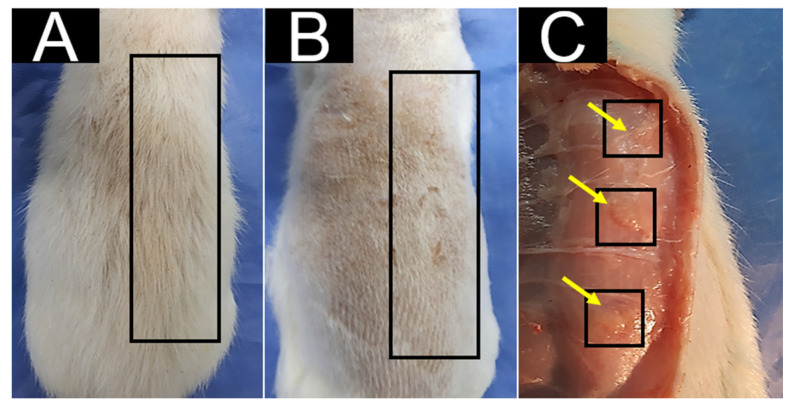
Macroscopic image of the dorsal area 30 days after implantation: (**A**) hair regrowth; (**B**) trichotomy area; (**C**) internal surface of the skin in the dorsal area. Rectangles: area of the dorsal surface where the procedure was performed. Squares: areas where the materials were implanted.

**Figure 7 polymers-15-03213-f007:**
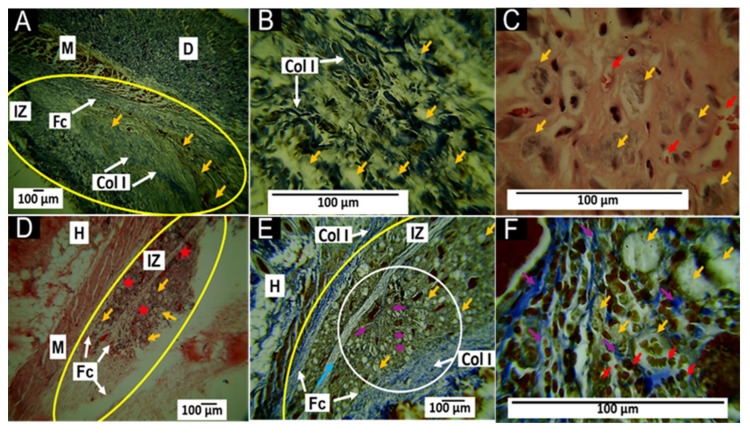
F1 formulation implanted at 30 and 60 days. (**A**) Implantation at 30 days, MT technique, 4× imaging; (**B**) implantation at 30 days, MT technique, 40× image; (**C**) 30-day implantation, HE technique, 100× imaging; (**D**) implantation at 60 days, HE technique, 4× imaging; (**E**) implantation at 60 days, MT technique, 10× imaging; (**F**) implantation at 60 days, MT technique, 100× imaging. D—dermis; M—muscle; IZ—implantation zone; FC—fibrous capsule; COL I—type I collagen fibers. Yellow arrows—particles of the material; red arrows—blood vessels; blue arrow—fibrous septum; red stars—inflammatory infiltrate; purple arrows—type I collagen fibers; yellow ovals—area of implantation; white circle—region of interest where collagen fibers were detached from the fibrous septum.

**Figure 8 polymers-15-03213-f008:**
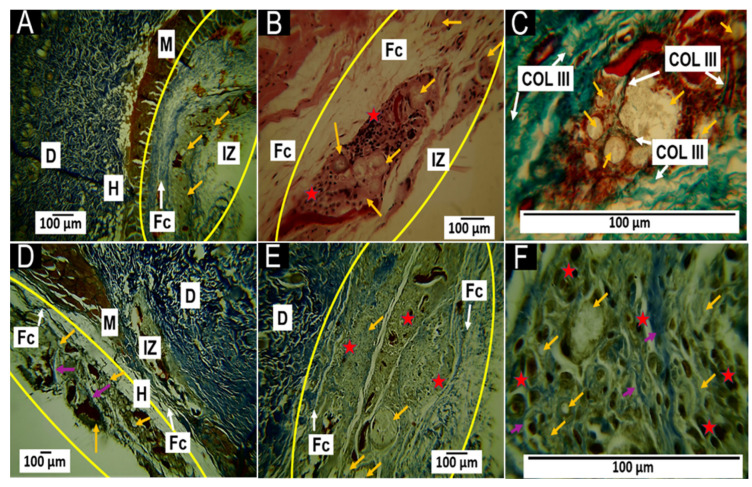
F2 formulation implanted at 30 and 60 days. (**A**) Implantation at 30 days, MT technique, 4× imaging; (**B**) implantation at 30 days, HE technique, 40× imaging; (**C**) 30-day implantation, GT technique, 100× imaging; (**D**) 60-day implantation, MT technique, 4× imaging; (**E**) 60-day implantation, MT technique, 10× imaging; (**F**) 60-day implantation, MT technique, 100× imaging. D—dermis; H—hypodermis; M—muscle; IZ—implantation zone; FC—fibrous capsule; COL III—type III collagen fibers. Yellow arrows—particles of the material; red stars—inflammatory infiltrate; purple arrows—type I collagen fibers; yellow ovals—implantation zone.

**Figure 9 polymers-15-03213-f009:**
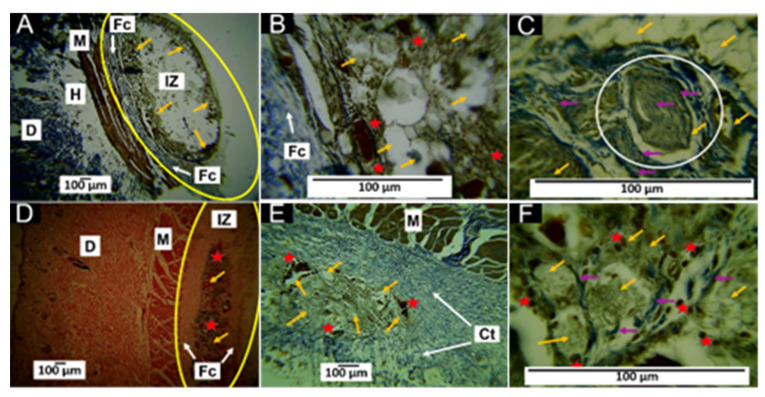
F3 formulation implanted at 30 and 60 days. (**A**) Implantation at 30 days, MT technique, 4× imaging; (**B**) implantation at 30 days, MT technique, 40× imaging; (**C**) 30-day implantation, MT technique, 100× imaging; (**D**) implantation at 60 days, HE technique, 4× imaging; (**E**) implantation at 60 days, MT technique, 10× imaging; (**F**) 60-day implantation, MT technique, 100× imaging. D—dermis; H—hypodermis; M—muscle; IZ—implantation zone; FC—fibrous capsule; Ct—connective tissue. Yellow arrows—particles of the material; red stars—inflammatory infiltrate; purple arrows—type I collagen fibers; yellow ovals—area of implantation. White circle: material covered by connective tissue.

**Table 1 polymers-15-03213-t001:** Composition of each of the components present in the CS/HA/β-TCP blocks.

Component	F1	F2	F3
β-TCP (%)	87.7	87.7	87.7
CS (%)	3.5	3.5	3.5
Commercial HA (%)	8.8	8.8	0
Natural HA (%)	0	0	8.8
5.4% CaCl_2_ (mL)	0.8	1.0	0.8

**Table 2 polymers-15-03213-t002:** EDS for the CS/HA/β-TCP composites. F1—87.7% β-TCP/3.5% CS/8.8% commercial HA/0.8 mL CaCl_2_; F2—87.7% β-TCP/3.5% CS/8.8% commercial HA/1 mL CaCl_2_; F3—87.7% β-TCP/3.5% CS/8.8% natural HA/0.8 mL CaCl_2_.

Sample	C (%)	O (%)	P (%)	Cl (%)	Ca (%)
F1	3.64 ± 4.08	51.57 ± 4.62	17.66 ± 2.11	0 ± 0	27.13 ± 1.72
F2	2.31 ± 4.01	53.29 ± 8.27	16.67 ± 1.34	0.51 ± 0.89	27.22 ± 3.58
F3	3.61 ± 4.18	48.98 ± 5.72	18.46 ± 1.33	0 ± 0	28.96 ± 2.86

## Data Availability

Data will be made available through request to the corresponding author.

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
