# Peer review of "Chitosan (CS)/Hydroxyapatite (HA)/Tricalcium Phosphate (β-TCP)-Based Composites as a Potential Material for Pulp Tissue Regeneration"

_polymers, 2023, doi:10.3390/polym15153213_

Round 1
Reviewer 1 Report
The manuscript "Chitosan (CS)/hydroxyapatite (HA)/tricalcium phosphate (β- TCP) based composites for tissue engineering applications" describes the preparation, physicochemical and biological characterisation of CS /HA/ β-TCP composites for intended use in dentistry. Although the work may be of interest to a wider audience, it has major and minor shortcomings that should be corrected before publication.
Major remarks
1. Introduction
The main flaw in the introduction is the lack of a description of previous work on CS /HA/ β-TCP. Although the authors claim that such ternary composites have not been studied, such studies exist, e.g. 10.1016/j.msec.2015.05.029, 10.1016/j.msec.2015.07.004., 10.3390/gels8110696.
Authors should expand the introduction to describe these previous studies and state the novelty of their work in relation to the previous studies.
2. Materials and methods
Some parts of the materials and methods are not described in sufficient detail to be reproduced. In more detail:
a) Equation 1 should be described.
b) An explanation should be given for the chosen composition of the composites, why such a small amount of HA and CS was chosen.
c)The resolution of the FTIR measurements should be given.
d)The step size and scan rate of the PXRD measurements should be given.
e) Explanation of the use of A. Sakina should be given.
f) A description of the statistical procedure should be given.
1. Results and discussion
The description and discussion of the FTIR and PXRD characterisation is not accurate, and the assignment is not always correct. Namely:
a) the band at 1020 cm-1 is also characteristic of ν3c triply degenerate asymmetric stretching mode of phosphate group (Koutsopoulos, S. Synthesis and Characterization of Hydroxyapatite Crystals: A Review Study on the Analytical Methods. J. Biomed. Mater. Res. 2002, 62, 600-612, Mochales, C.; Wilson, R.M.; Dowker, S.E.P.; Ginebra, M.-P. Dry Mechanosynthesis of Nanocrystalline Calcium Deficient Hydroxyapatite: Structural Characterisation. J. Alloys Compd. 2011, 509, 7389-7394). Although the vibration at 1020 cm-1 is characteristic of C=O vibration in CS (Tanase, C.E.; Popa, M.I.; Verestiuc, L. Biomimetic Chitosan-Calcium Phosphate Composites with Potential Applications as Bone Substitutes: Preparation and Characterization. J. Biomed. Mater. Res. 2012, 100B, 700-708.), considering that the total percentage of HA and β-TCP in the composites is 96.5%, this band cannot be assigned to the C=O vibration. It is true that it is challenging to assign the CS bands unambiguously because the most intense CS bands overlap with phosphate bands.
b) The discussion of the FTIR spectra (lines 207-214) is not clear. Were the composites calcinated? Calcination has only been described in the synthesis of natural HA. Why then do the authors state that "the bands observed in the spectrum are mainly due to the calcination process"? Since all 3 spectra are similar, calcination should not play a role.
c) I have a problem with the assignment of the β-TCP reflections. Also, the reference given does not deal with β-TCP. The authors should assign the reflexes according to the corresponding powder diffraction card. Also, the 2q values should be given with more precision.
d) The HA reflexes should also be assigned according to the corresponding powder diffraction card.
e) The discussion in lines 230-236 is not clear. The reflection at about 2q 11° could be assigned to HA or β-TCP (100).
To make it entirely accurate, the FTIR spectra and PXRD patterns of starting components, both HA, β-TCP and CS should be given. The assignation should be made using correct references and/or powder diffraction cards.
The discussion of the SEM results (lines 238-247) also focuses on the influence of calcination. However, it was not stated that the composites were calcined. Crystallinity is also mentioned, but it is not shown how it was calculated. It is also not clear why "the crystallinity of the composites, as shown in the XRD patterns in Figure 1, indicates the formation of more than one phase, probably due to a large size distribution."
Furthermore, since the presence of CS was not clearly confirmed, the authors should carry out a TGA analysis.
Minor remarks
1. Abstract
- Instead, "The chemical characterization using infrared (FTIR) revealed the presence of PO4 functional groups in both HA and β-TCP. X-ray diffraction (XRD) analysis identified the different crystal forms of β-TCP, CS, and HA, with β-TCP showing a higher prevalence due to the intensity of the (201) planes."
it would suffice to say that FTIR and PXRD characterisation confirmed the composition of the composites.
- The last sentence mentions infection conditions, although no evidence of the anti-infective activity of investigated composites is given in the paper.
2. Introduction
- Ref 2 should be replaced by a review paper dealing with scaffolds in regenerative dentistry.
- HA can be prepared by various methods, not only by a precipitation reaction of Ca(OH)2 and H3PO4. Please correct.
- Ref 15 and 16 should be replaced with a more recent articles.
- CS is a polysaccharide and not a cationic polymer.
3. Materials and methods
- Lines 165-171 should be corrected to avoid repetition of information.
- Ref. 28 does not deal with the procedures for obtaining histological sections.
4. Results and discussion
- Line 193 - "y" should be changed to "- "
- Ref. 30 does not deal with FTIR spectra of CS.
- A more appropriate reference for HA PXRD patterns should be given instead of Ref. 34.
- The results of the EDS analysis could be included in Figure 3. or as a separate table to avoid repeating the SEM images in Figure SI 1. The average content of elements and SD should also be given.
- Table 6. - the SD values for dead larvae and percentage mortality should be given. The number of initial A. salina larvae is redundant data; it could be explained in the table caption. In addition, the table should be labeled 2.
- Figure 5 and Table 6 repeat the same data. The authors should decide which to use.
I would suggest the correction of the English language, as some sentences are not clear.
Author Response
Reviewer 1 Major remarks 1. Introduction The main flaw in the introduction is the lack of a description of previous work on CS /HA/ β-TCP. Although the authors claim that such ternary composites have not been studied, such studies exist, e.g. 10.1016/j.msec.2015.05.029, 10.1016/j.msec.2015.07.004., 10.3390/gels8110696. Authors should expand the introduction to describe these previous studies and state the novelty of their work about the previous studies. R// Thank you for your appreciation. You can find the correction from lines 79-95 in the manuscript. In this sense, CS has been blended with different synthetic polymers or mineralized salts to compensate for these disadvantages, such as the ternary mixture between CS/HA/β-TCP, where it was shown that the crosslinking process influences its physicochemical and mechanical properties. In addition, this composite showed a more significant interaction between HA/β-TCP due to the presence of CS, which facilitated a uniform distribution in the scaffold [1]. Additionally, this ternary blend has shown good swelling, mechanical and thermal properties related to the HA/ β -TCP ratio. Cell culture experiments showed that L929 and Saos-2 cells increased on the surface of the composite scaffold and that a higher HA/β-TCP ratio stimulated cell growth [2]. To date, there are no studies of subdermal implantation according to the biocompatibility of the material attributed to the ternary mixture CS/HA/β-TCP taking into account the possible variations in the physicochemical properties due to the implementation of natural or synthetic HA. The results of the in vitro and in vivo biocompatibility evaluation demonstrated the potential of these materials in the field of tissue engineering due to the emergence of connective tissue and the decrease of inflammatory response. 2. Materials and methods Some parts of the materials and methods are not described in sufficient detail to be reproduced. In more detail: a) Equation 1 should be described. R// Thank you for your appreciation. We added information as follows: (lines 102-115). A capillary viscometer determined the molecular weight of CS, and gel permeation chromatography (GPC) used viscous solutions of CS, allowing the measurement of the average viscous molecular weight (Mv). For this purpose, the drop times of 25 mL solutions were measured with a Ubbelohde viscometer prepared from a standard solution (Fungilab, Santiago de Chile, Chile). The intrinsic viscosity is the intercept value of the specific viscosity curve and is symbolized by [η]. The equation Mark Houwink-Sakurada (equation 1) relates this viscosity to the constants K (0.074 mL/g) and a (0.76) estimated at 25°C in a mixture of 0.3 M acetic acid and 0.2 M sodium acetate [3], where the Mv calculated for the CS was of 144000 Da. b) An explanation should be given for the chosen composition of the composites, why such a small amount of HA and CS was chosen. R// Thank you very much for your suggestion. The correction can be found between lines 133-136. To obtain a composite material with good structural integrity and sound osteoconductive reinforcement, small amounts of HA were taken not to affect the composite's resorption processes, and small parts of CS were not to induce strong inflammatory responses. c) The resolution of the FTIR measurements should be given. R// Thank you for your observation. The resolution of the measurement in the FT-IR was 4 cm-1. Line 148. The determination of the individual functional groups present in the blocks CS/HA/β-TCP was performed by FT-IR on an Affinity-1 IR infrared spectrophotometer (Shimadzu, Kyoto, Japan) in the range 500-4000 cm-1 with 4 cm-1 spectral resolution in transmittance mode using the attenuated total reflectance (ATR) method with a diamond tip. d) The step size and scan rate of the PXRD measurements should be given. R// Thank you for your appreciation. The measurement was performed at a scan rate of 2.63 s. Line 157. The presence of the different crystalline planes in the CS/HA/β-TCP blocks was determined employing a PANalytical X'Pert PRO MRD diffractometer (Malvern Panalytical, Jarman Way, Royston, UK) using copper radiation at a wavelength of Kα1 (1.540598 Å) and Kα2 (1.544426 Å) operated in the secondary electron mode at 45 kV in the range 2θ between 5 and 80° with a scan speed of 2.63 s. e) Explanation of the use of A. Salina should be given. R// Thank you very much for the suggestion. The correction can be found as follows: (lines 171-172). A. Salina is a test that allows a preliminary approximation of the material's toxicity, which is standardized and recognized in the literature. f) A description of the statistical procedure should be given. R// Thank you for your observation. The description of the statistical procedure is seen in the final section of materials and methods for details (Lines 210-214). Biocompatibility results for interaction with A. salina have presented as the mean value of at least five replicates ± SD. Analysis of variance (ANOVA) and Tukey's method at 95% confidence level (α = 0.05) were used to assess the significance of changing formulations on the in vitro characterization responses of the blocks. Statgraphics Centurion XVI was used for statistical analyses. 1. Results and discussion The description and discussion of the FTIR and PXRD characterization is not accurate, and the assignment is not always correct. Namely: a) the band at 1020 cm-1 is also characteristic of ν3c triply degenerate asymmetric stretching mode of phosphate group (Koutsopoulos, S. Synthesis and Characterization of Hydroxyapatite Crystals: A Review Study on the Analytical Methods. J. Biomed. Mater. Res. 2002, 62, 600-612, Mochales, C.; Wilson, R.M.; Dowker, S.E.P.; Ginebra, M.-P. Dry Mechanosynthesis of Nanocrystalline Calcium Deficient Hydroxyapatite: Structural Characterisation. J. Alloys Compd. 2011, 509, 7389-7394). Although the vibration at 1020 cm-1 is characteristic of C=O vibration in CS (Tanase, C.E.; Popa, M.I.; Verestiuc, L. Biomimetic Chitosan-Calcium Phosphate Composites with Potential Applications as Bone Substitutes: Preparation and Characterization. J. Biomed. Mater. Res. 2012, 100B, 700-708.), considering that the total percentage of HA and β-TCP in the composites is 96.5%, this band cannot be assigned to the C=O vibration. It is true that it is challenging to assign the CS bands unambiguously because the most intense CS bands overlap with phosphate bands. R// Thank you very much for your suggestion. The correction can be found as follows: (lines 224-226). The band located at 1020 cm-1 overlaps with the triple-degenerate asymmetric stretching of the PO4 group in hydroxyapatite [4] and the vibrational mode of the C-O-C bond in chitosan [5]. b) The discussion of the FTIR spectra (lines 207-214) is not clear. Were the composites calcinated? Calcination has only been described in the synthesis of natural HA. Why then do the authors state that "the bands observed in the spectrum are mainly due to the calcination process"? Since all 3 spectra are similar, calcination should not play a role. R// Thank you very much for the suggestion. Re-analyzing the FT-IR spectra and clarifying that the calcination was carried out only for natural HA, we deleted from the paper the discussion on how the calcination process affects the composition of the material and added the following: (lines 240-245). The FTIR spectra of all three formulations indicate that the characteristic bands of CaP and CS are present in the composites. Additionally, these results suggest that the origin of the HA does not matter because no significant differences are found in the FT-IR spectra of the three formulations. Therefore, having the same weight radius within the composites, no changes in the intensities of the PO4 or CS groups are observed as in previous research [1,2]. c) I have a problem with the assignment of the β-TCP reflections. Also, the reference given does not deal with β-TCP. The authors should assign the reflexes according to the corresponding powder diffraction card. Also, the 2q values should be given with more precision. R// Thank you very much for the suggestion. We corrected the text in the discussion of XRD as follows: (lines 248-258). Figure 2 shows the interaction between the X-rays and the composites to give rise to the characteristic crystalline planes of each component. In the diffractogram, the displacements 2θ characteristics of the β-TCP structure were observed, which present peaks at 21.8, 25.8, 27.8, 29.6, 31.0, 32.5, 34.4, 35.6, 39.8, 47.0, 48.0, and 53° related to planes (024), (1010), (214), (300), (0210), (128), (220), (128), (1016), (4010), (238), and (2020), the latter being the most representative to confirm a tetragonal phase with a 76-P41 space group indexed through research previous [6,7]. Nevertheless, crystalline planes of the HA were found to overlap on the characteristic plane of β-TCP. In this sense, we can find plans (010), (011), (110), (020), (111), (021), (002), (012), (120), (121), among others, which are consistent with the reference code 98-010-5020 for a hexagonal structure with space group P 63/m. d) The HA reflexes should also be assigned according to the corresponding powder diffraction card. R// Thank you very much for your appreciation. As with the previous question, the assignment of the plans has been corrected. Lines 248-258. e) The discussion in lines 230-236 is not clear. The reflection at about 2q 11° could be assigned to HA or β-TCP (100). R// Thank you very much for the observation. The corresponding section added the following text: (lines 263-269). On the other hand, previous studies have reported six polymorphisms for the chitosan "tendon chitosan," "annealed," "1-2", "L-2", "form-I," and "form-II." Furthermore, for all formulations, 2θ shifts at 10 and 20° were observed, attributed to the (122) and (110) planes with semi-crystalline nature related [8]. However, in these results, the crystallinity of CS is affected by the presence of HA/β-TCP because this bicomponent mixture weakens the intramolecular interactions of the chitosan chain [9]. This observation is similar to other previously reported works [10–12]. The discussion of the SEM results (lines 238-247) also focuses on the influence of calcination. However, it was not stated that the composites were calcined. Crystallinity is also mentioned, but it is not shown how it was calculated. It is also not clear why "the crystallinity of the composites, as shown in the XRD patterns in Figure 1, indicates the formation of more than one phase, probably due to a large size distribution." R// Thank you very much for the suggestion. The following text was added: (lines 272-282). Micrographic studies of the CS/HA/β-TCP composites are shown in Figure 3. The surface micrographs show an agglomerated and spherical structure. In general, the grain size for such composites depends on the calcination temperature at which they are prepared [6,7]. However, in our case, as the material was not calcined, no trend in grain size was found. Therefore, it can be said that the composition and origin of the HA, whether natural or commercial, do not affect the morphology of the composites. Additionally, previous studies of HA/β-TCP composites as a function of their periodicity showed that the increase of bilayers in the membrane does not affect the overall thickness of the total material, causing a decrease in the thickness of the individual layers [13], which probably suggests in our case that the increase in grain size depends on whether the composites are made at different temperatures [6]. Minor remarks 1. Abstract - Instead, "The chemical characterization using infrared (FTIR) revealed the presence of PO4 functional groups in both HA and β-TCP. X-ray diffraction (XRD) analysis identified the different crystal forms of β-TCP, CS, and HA, with β-TCP showing a higher prevalence due to the intensity of the (201) planes." it would suffice to say that FTIR and PXRD characterisation confirmed the composition of the composites. R// Thank you very much for your comment. The suggestion was corrected as follows: (lines 22-23). The chemical characterization using infrared spectroscopy (FTIR) and X-ray diffraction (XRD) analysis confirmed the composition of the composites. - The last sentence mentions infection conditions, although no evidence of the anti-infective activity of investigated composites is given in the paper. R// Thank you very much for your comment. The suggestion was corrected as follows: (lines 31-32). The results obtained in this research contribute to developing endodontic technologies for tissue recovery and regeneration. 2. Introduction - Ref 2 should be replaced by a review paper dealing with scaffolds in regenerative dentistry. R// Thank you for your suggestion. We changed the reference to: Amrollahi, P.; Shah, B.; Seifi, A.; Tayebi, L. Recent advancements in regenerative dentistry: A review. Mater. Sci. Eng. C 2016, 69, 1383–1390 - HA can be prepared by various methods, not only by a precipitation reaction of Ca(OH)2 and H3PO4. Please correct. R// Thank you for your observation. The text was corrected as follows: (lines 62-65). HA is synthesized through four different methods: i) dry methods, ii) wet methods, iii) microwave (MW) assisted method, ball-milling or ultrasound, and iv) various methods [14]. - Ref 15 and 16 should be replaced with a more recent article. R// Thank you very much for the suggestion. Reference 15, which in the new text is 14, was changed to the following reference: "Noordin, N.N.F.N.M.; Ahmad, N.; Mariatti, M.; Yahaya, B.H.; Sulaiman, A.R.; Hamid, Z.A.A. A review on bioceramics scaffolds for bone defect in different animal models: HA and β-TCP. Biomed. Phys. Express 2022". Also, reference 16, which in the new text is 15, was changed to "Helaehil, J.V.; Lourenço, C.B.; Huang, B.; Helaehil, L.V.; de Camargo, I.X.; Chiarotto, G.B.; Santamaria-Jr, M. Bártolo, P.; Caetano, G.F. In vivo investigation of polymer-ceramic PCL/HA and PCL/β-TCP 3D composite scaffolds and electrical stimulation for bone regeneration. Polymers (Basel). 2021, 14, 65". - CS is a polysaccharide and not a cationic polymer. R// Thank you very much for the observation. The correction of the type of CS molecule has been added in line 72. 3. Materials and methods -Lines 165-171 should be corrected to avoid repetition of information. R// Thank you very much for the suggestion. The decision was taken to summarise a large part of the protocol, limiting it to the explanation after implantation because the subdermal fixation protocol must be carried out considering the iso 10993-6 standard. (Lines 188-192). For the in vivo biocompatibility test, six male Wistar rats (Rattus norvegicus domestic), five months old, average weight of 380 grams, were randomly distributed in two groups for observations at 30 and 60 days, taking into account that the minimum number of rats to obtain results should be three according to the ISO 10993-6 standard [15], which made it possible to apply Rusell and Burch's principle of the three Rs - Ref. 28 does not deal with the procedures for obtaining histological sections. R// Thank you very much for the comment. The two references were changed and are seen in the text as [29,30] in the following way: "Grande-Tovar, C.D.; Castro, J.I.; Valencia Llano, C.H.; Tenorio, D.L.; Saavedra, M.; Zapata, P.A.; Chaur, M.N. Polycaprolactone (PCL)-Polylactic acid (PLA)-Glycerol (Gly) Composites Incorporated with Zinc Oxide Nanoparticles (ZnO-NPs) and Tea Tree Essential Oil (TTEO) for Tissue Engineering Applications. Pharmaceutics 2023, 15, 43”. and “Castro, J.I.; Valencia-llano, C.H.; Eliana, M.; Zapata, V.; Restrepo, Y.J.; Mina, H.; Navia-porras, D.P.; Valencia, Y.; Valencia, C.; Grande-tovar, C.D. Chitosan / Polyvinyl Alcohol / Tea Tree Essential Oil Composite Films for Biomedical Applications. 2021, 1-21". 4. Results and discusión - Line 193 - "y" should be changed to "-. " R// Thank you very much for the suggestion. The suggested change was incorporated as follows: line 222 In this context, the stress vibrational modes of the PO_4^(-3) bond is at 936 and 1100 cm-1 - Ref. 30 does not deal with FTIR spectra of CS. R// Thank you very much for your comments. We changed the reference to "Tanase, C.E.; Popa, M.I.; Verestiuc, L. Biomimetic chitosan-calcium phosphate composites with potential applications as bone substitutes: Preparation and characterization. J. Biomed. Mater. Res. - Part B Appl. Biomater. 2012, 100 B, 700–708, doi:10.1002/jbm.b.32502". - A more appropriate reference for HA PXRD patterns should be given instead of Ref. 34. R// Thank you very much for the comment. Due to the new discussion of results by the PXRD, reference 34 has been deleted. - The results of the EDS analysis could be included in Figure 3 or as a separate table to avoid repeating the SEM images in Figure SI 1. The average content of elements and SD should also be given. R// Thank you very much for the suggestion. The table was included in the discussion of the results. Line 297. - Table 6. - the SD values for dead larvae and percentage mortality should be given. The number of initial A. salina larvae is redundant data; it could be explained in the table caption. In addition, the table should be labeled 2. R// Thank you very much for the suggestion. Based on what was requested, it was decided to remove the table and leave only Figure 5. - Figure 5 and Table 6 repeat the same data. The authors should decide which to use. R// Thank you very much for the suggestion. Due to the above correction, the table number is not considered, as the table was removed from the manuscript.

Reviewer 2 Report
Thanks for considering me as a reviewer for this valuable study. I reviewed it and have some comments.
Indeed, Zamora et al studied a new composite "Chitosan (CS)/hydroxyapatite (HA)/tricalcium phosphate (β- 2 TCP)' for tissue engineering applications.
First, the title is so general. Which type of tissue engineering? I noticed that the author focused on The pulp tissue of teeth in the introduction section. Then, in my opinion, it should be reflected in the title. besides, all used materials in the composite structures are dental biomaterials.
The conclusion of the abstract is not so proper. It does not support the results. Please improve.
The introduction provides sufficient background and includes all relevant references. However, some recent and potent references are needed to add. Besides, some sections of the introduction need a reference (s). Please check.
The research design is appropriate.
The methods are adequately described. However, for a composite, I think The TEM image is necessary. Please add or discuss.
Discussion is potent. However, I think there are some recent studies in this regard that can be added and compared with the current studies' results.
Please add study limitations and the future perspectives for this new composite for clinical use in endodontics.
Minor editing of English language required.
Author Response
Reviewer 2
Indeed, Zamora et al studied a new composite "Chitosan (CS)/hydroxyapatite (HA)/tricalcium phosphate (β- 2 TCP)' for tissue engineering applications.
R// Thank you very much for the observation. Several studies involve the tricomponent mixture CS/HA/β- TCP [1,2]. However, no studies concerning the subdermal evaluation of the component provide applied science in tissue engineering with a focus on pulp regeneration.
First, the title is so general. Which type of tissue engineering? I noticed that the author focused on The pulp tissue of teeth in the introduction section. Then, in my opinion, it should be reflected in the title. Besides, all used materials in the composite structures are dental biomaterials.
R// Thank you very much for your suggestion. The new title added was: "Chitosan (CS)/hydroxyapatite (HA)/tricalcium phosphate (β-TCP) based composites as a possible material to be applied in pulp tissue regeneration."
The conclusion of the abstract is not so proper. It does not support the results. Please improve.
R// Thank you very much for the observation. The conclusions and the summary were appropriately corrected.
The introduction provides sufficient background and includes all relevant references. However, some recent and potent references are needed to add. Besides, some sections of the introduction need a reference (s). Please check.
R// Thank you very much for your appreciation. In the introduction, we added references to studies that have contained the triphasic mixture applied to tissue engineering and reviews in the field of endodontics.
"Amrollahi, P.; Shah, B.; Seifi, A.; Tayebi, L. Recent advancements in regenerative dentistry: A review. Mater. Sci. Eng. C 2016, 69, 1383–1390". Reference 2
"Fihri, A.; Len, C.; Varma, R.S.; Solhy, A. Hydroxyapatite: A review of syntheses, structure, and applications in heterogeneous catalysis. Coord. Chem. Rev. 2017, 347, 48–76, doi:10.1016/j.ccr.2017.06.009”. Reference 10
"Shavandi, A.; Bekhit, A.E.D.A.; Sun, Z.; Ali, A.; Gould, M. A novel squid pen chitosan/hydroxyapatite/β-tricalcium phosphate composite for bone tissue engineering. Mater. Sci. Eng. C 2015, 55, 373–383, doi:10.1016/j.msec.2015.05.029". Reference 21
"Shavandi, A.; Bekhit, A.E.D.A.; Ali, M.A.; Sun, Z.; Gould, M. Development and characterization of hydroxyapatite/β-TCP/chitosan composites for tissue engineering applications. Mater. Sci. Eng. C 2015, 56, 481–493, doi:10.1016/j.msec.2015.07.004". Reference 22
The research design is appropriate.
R// Thank you very much for your appreciation.
The methods are adequately described. However, for a composite, I think The TEM image is necessary. Please add or discuss.
R// We appreciate the valuable suggestion. However, we believe that a morphological analysis is sufficient to demonstrate the presence of pores and their complement with EDS to have a first approximation of the percentage composition of the scaffold. On the other hand, a TEM analysis is performed to analyze small dimensionality because transmission microscopy allows a higher resolution power to achieve three-dimensional images. For the study, we considered that running a TEM is unnecessary. However, in the future, we can consider doing TEM analysis for a higher resolution in upcoming studies.
Discussion is potent. However, I think some recent studies in this regard can be added and compared with the current studies' results.
R// Thank you very much for the observation. The results have been complemented by relevant studies in the field, especially in the section on the characterization of the composites, also taking into account the suggestions of Referee 1.
Please add study limitations and the future perspectives for this new composite for clinical use in endodontics.
R// Thank you very much for the suggestion. The text was added as follows: (lines 495-500).
Given the above, these approximations in the behavior of the different formulations give an idea of the human body's process once the blocks have been implanted in subdermal tissue for their regenerative evolution, which resembles pulp tissue. However, the in vitro study employing A. Salina and subdermal biocompatibility using biomodels needs complementary cytotoxicity studies with pulp cells and in vivo tests using a dental animal model to approach the pulp object of study.

Reviewer 3 Report
The introduction, despite its considerable length, contains several controversial statements.
Line 60-62: several TCP-precipitation techniques are known, as the several solid-state processes of HA preparation. Moreover, Ca(NO3)2 and (NH4)2HPO4 are the most common reagents for HA precipitation, neither Ca(OH)2 and H3PO4.
The authors did not indicate how the solubility of TCP and HA correlates with the solubility of the chitosan matrix.
In general, the introduction of a two-level (by solubility) mineral component is not substantiated, nor is the HA/TCP/CS ratio in substantiated.
Materials
One of the significant problems in the use of chitosan is its purity, namely, the impurities of chitin from which it is obtained. The authors do not indicate the purity qualification of chitosan, it is not clear whether they carried out additional purification of chitosan, considering its use as a biomaterial.
The choice of chitosan with just such a molecular weight is not justified.
Results
Discussion of the XRD results seems to be controversial. It was necessary to present the diffraction patterns of the individual components of the composite to make correct conclusions about the peaks superposition. The same relates to FTIR.
Microstructure (Fig.3) of composites seems to almost the same for all samples.
Fig.4 poor quality.
Line 52 double MTA abbreviature introduction
Author Response
Reviewer 3
The introduction, despite its considerable length, contains several controversial statements.
R// Thank you very much for the suggestion. The introduction was improved according to all the recommendations made by the reviewers.
Line 60-62: several TCP-precipitation techniques are known, as the several solid-state processes of HA preparation. Moreover, Ca(NO3)2 and (NH4)2HPO4 are the most common reagents for HA precipitation, neither Ca(OH)2 and H3PO4.
R// Thank you very much for the suggestion. The methods for obtaining the components are detailed in the following text with their respective reference (Lines 60-63).
The solid-state reaction by thermal decomposition generally prepares the β-TCP powders. At the same time, HA is synthesized through four different methods: i) dry methods, ii) wet methods, iii) microwave (MW) assisted method, ball-milling or ultrasound, and iv) various methods [14].
The authors did not indicate how the solubility of TCP and HA correlates with the solubility of the chitosan matrix.
R// Thank you very much for your comment. According to the above suggestions, we were able to investigate how the CS acts as a support for the distribution of the components on the polymeric matrix. For this purpose, the following paragraph was corrected: (lines 79-84).
In this sense, CS has been blended with different synthetic polymers or mineralized salts to compensate for these disadvantages, such as the ternary mixture between CS/HA/β-TCP, where it was shown that the crosslinking process influences its physicochemical and mechanical properties. In addition, this composite showed a more significant interaction between HA/β-TCP due to the presence of CS, which facilitated a uniform distribution in the scaffold [1].
In general, the introduction of a two-level (by solubility) mineral component is not substantiated, nor is the HA/TCP/CS ratio in substantiated.
R// Thank you very much for your appreciation. The introduction was intended to demonstrate a gap in the field of pulp tissue regeneration. However, different materials have been used, such as MTA, which has proven not to be very efficient in its degradability due to the presence of some components after a long time. For this reason, the formation of mineralized mixtures that promote the regeneration of pulp tissue and increased resorption without necrotic tissue has been proposed. In this respect, combinations between β-TCP and HA have been explored as future components with the help of natural compounds with CS, which have good biological properties and allow good adhesion with other structures in general. For this reason, it is said that the formation of composites containing CS/β-TCP/HA could serve as materials that promote pulp tissue regeneration without necrotic processes.
Materials
One of the significant problems in the use of chitosan is its purity, namely, the impurities of chitin from which it is obtained. The authors do not indicate the purity qualification of chitosan; it is not clear whether they carried out additional purification of chitosan, considering its use as a biomaterial.
R// Thank you very much for the suggestion. The materials and methods section specified that the chitosan used was purchased from a commercial company, which guarantees that its degree of deacetylation is between 89-90%. For this reason, no additional purification methods were used; therefore, it was used directly according to the degree of incorporation for each formulation. Lines 112-115.
The degree of CS deacetylation was 89-90% and was determined by 400 MHz 1H-NMR spectroscopy using a BRUKER AVANCE II (Bruker, Berlin, Germany) with 300 K temperature control and a Thermo Electron Flash EA 1112 elemental analyzer (Thermo Fischer, Waltham, MA, USA).
The choice of chitosan with just such a molecular weight is not justified.
R// Thank you very much for the observation. The choice of chitosan was made according to previous research in the research group [16–18], as it has allowed it to be one of the components that promote the regeneration of tissue without the presence of pathogenic or necrotic agents that prevent its use for this type of application.
Results
Discussion of the XRD results seems to be controversial. It was necessary to present the diffraction patterns of the individual components of the composite to make correct conclusions about the peaks superposition. The same relates to FTIR.
R// Thank you very much for the suggestion. The remarks made by the referees are related to the assignment of the diffraction peaks, which were made according to the charts and articles mentioning the exact position of the sample to be analyzed. On the other hand, it can be observed that in the FTIR, it is challenging to assign the corresponding band for each component due to the overlapping of the bands and the slight variation in weight, which probably suggests that commercial HA as a natural implementation does not differ significantly concerning the composite structure formed by the three components.
Microstructure (Fig.3) of composites seems to almost the same for all samples.
R// Thank you very much for the observation. The microstructures are similar because they use the same percentage by weight of each component. However, the implementation of natural or synthetic HA varies, which suggests that no significant differences are found in their morphology concerning the type of origin of the HA.
Fig.4 poor quality.
R// Thank you very much for your appreciation. The image quality has been improved.
Line 52 double MTA abbreviature introduction
R// Thank you very much for the observation. We corrected the text as follows: lines 52-54.
The MTA consists of calcium, aluminum, and selenium, which has become the most studied endodontic material in the last two decades due to its biocompatibility, hydrophilicity, bioactivity, radio-opacity, sealing capacity, and low solubility [19,20].

Round 2
Reviewer 1 Report
The authors have responded to the majority of the remarks, however, there are some ambiguities left that need to be resolved before the paper could be accepted for publishing.
Major:
There is still ambiguity in the PXRD characterization of mineral phases. The obtained patterns should be assigned according to corresponding ICCD cards, not only according to previous papers.
Minor:
- The title can be simplified to "...based composites as a potential material for pulp tissue regeneration".
- The step size and scan rate of the PXRD measurements should be properly given.
- The reference for Mark Houwink-Sakurada equation should be given.
- Molecular weight should be designated as Mw, not Mv (lines 103 and 109)
- in lines 261 - 265 it is stated:
"Additionally, previous studies of HA/β-TCP composites as a function of their periodicity showed that the increase of bilayers in the membrane does not affect the overall thickness of the total material, causing a decrease in the thickness of the individual layers [42], which probably suggests in our case that the increase in grain size depends on whether the composites are made at different temperatures [35]."
First, it is not clear which membranes the authors are talking about and why.
Second, the samples were not calcinated, therefore how can the change in grain size depend on different temperatures?
- The SI is not needed anymore.
- English language should be thoroughly checked as there are some instances in which it is not clear what the authors wanted to say. Some examples are:
- what is weight radius in line 230
- please correct the bold parts in the following statements:
1. lines 233-235: Figure 2 shows the interaction between the X-rays and the composites to give rise to the characteristic crystalline planes of each component. In the diffractogram, the displacements 2θ characteristics ...
2. lines 239 - 240 crystalline planes of the HA were found to overlap on the characteristic plane of β-TCP.
Comments are given in comments for the authors.
Author Response
We appreciate all the suggestions from the reviewer. Here are the answers, point by point, to each comment:
Reviewer 1
Major:
There is still ambiguity in the PXRD characterization of mineral phases. The obtained patterns should be assigned according to corresponding ICCD cards, not only according to previous papers.
R// Thank you for your appreciation. You can find the correction from lines 249-259 in the manuscript.
Figure 2 shows the XRD pattern of the blocks of CS/HA/β-TCP. In the diffractogram, the characteristic diffraction peaks of the β-TCP structure were observed, which present angles 2θ at 13.6, 17.0, 21.8, 25.7, 26.5, 27.8, 29.6, 32.4, 35.6, 39.8, 41.1, and 41.7° related to reflections of the crystallographic planes (004), (110), (113), (106), (200), (202), (008), (205), (215), (217), (224), and (303), the latter being the most representative to confirm a tetragonal phase with a 76-P41 space group, according the international files JCPDF 00-020-0024 [1]. Additionally, it was observed that some HA angles overlap with those belonging to the β-TCP. In this sense, it was possible to follow the angles 2θ at 10.8, 21.8, 31.0, 34.4, 35.6, 47.0, 48.0, 48.4, 51.5, and 53.0° attributed to the crystallographic planes (010), (020), (121), (022), (031), (222), (132), (230), (140), and (004), which are consistent with the reference code ICCD 98-010-5020 for a hexagonal structure with space group P 63/m.
Minor:
- The title can be simplified to "...based composites as a potential material for pulp tissue regeneration".
R// Thank you for your appreciation. We modify the title as follows:
Chitosan (CS)/hydroxyapatite (HA)/tricalcium phosphate (β-TCP) based composites as a potential material for pulp tissue regeneration.
- The step size and scan rate of the PXRD measurements should be properly given.
R// Thank you for your appreciation. The measurement was performed on the manuscript, lines 156-157.
Operated in the secondary electron mode at 45 kV in the range 2θ between 5 and 80° at a scan rate of 2 degrees/min with a scan speed of 2.63 s and a step size of 0.01°.
-The reference for Mark Houwink-Sakurada equation should be given.
R// Thank you for your observation. The reference is highlighted in line 107, which corresponds to:
Kasaai, M.R. Calculation of Mark–Houwink–Sakurada (MHS) equation viscometric constants for chitosan in any solvent–temperature system using experimental reported viscometric constants data. 2007, 68, 477–488, doi:10.1016/j.carbpol.2006.11.006.
- Molecular weight should be designated as Mw, not Mv (lines 103 and 109)
R// Thank you very much for the suggestion. However, we disagree with this correction because the molecular weight calculated by the Mark-Houwink-Sakurada equation is related to the intrinsic viscosity of the polymer and the average molecular weight. The latter is internationally known by the acronym "Mv" because it is based on measuring the numerical molecular weight considering the polymer's intrinsic viscosity. Intrinsic viscosity measures the hydrodynamic volume occupied by macromolecules in solution and, therefore, reflects their size; [η] is highly dependent on the polymer concentration in solution. Experimentally, [η] is obtained from relative viscosity measurements of solutions with different polymer concentrations. The measurements are performed in a capillary viscometer, which can be of the Ubbelohde, Ostwald, or Cannon-Fenske type, at a given temperature, based on Poiseuille's law and assuming Newtonian behavior for low concentrations and low shear forces. The value of [η] is obtained by extrapolating η to zero concentration [2].
We only clarify the expression in the highlighted lines, but we cannot add Mw as this acronym refers to molecular mass.
- in lines 261 - 265 it is stated:
"Additionally, previous studies of HA/β-TCP composites as a function of their periodicity showed that the increase of bilayers in the membrane does not affect the overall thickness of the total material, causing a decrease in the thickness of the individual layers [42], which probably suggests in our case that the increase in grain size depends on whether the composites are made at different temperatures [35]."
First, it is not clear which membranes the authors are talking about and why.
Second, the samples were not calcinated, therefore how can the change in grain size depend on different temperatures?
R// Thank you very much for the observation. In previous work with the same components, the porosity and roughness depend mainly on the temperature and weight percentage of each element present in the composite. Therefore, as in our case, the samples were not calcined and kept the same weight ratio for each formulation; the morphology was unaffected.
In this sense, it was decided to eliminate this section to avoid inconsistencies during the discussion of the results.
- The SI is not needed anymore.
R// Thank you very much for the comment. The SI will be removed.
- English language should be thoroughly checked as there are some instances in which it is not clear what the authors wanted to say. Some examples are:
- what is the weight radius in line 230
R// Thank you very much for the suggestion. The phrase weight ratio was replaced by weight percentage. Line 245. The English language was thoroughly reviewed across the manuscript.
- please correct the bold parts in the following statements:
- lines 233-235: Figure 2 shows the interaction between the X-rays and the composites to give rise to the characteristic crystalline planes of each component. In the diffractogram, the displacements 2θ characteristics.
R// Thank you very much for the suggestion. We removed the bold in the text (249-251):
Figure 2 shows the XRD pattern of the blocks of CS/HA/β-TCP. The characteristic diffraction peaks of the β-TCP structure were observed in the diffractogram, which presents angles.
- lines 239 - 240 crystalline planes of the HA were found to overlap on the characteristic plane of β-TCP.
R// Thank you very much for the suggestion. The bold was also removed from the text (lines 255-256):
Additionally, it was observed that some HA angles overlap with those belonging to the β-TCP. In this sense, it was possible to follow the angles 2θ at 10.8, 21.8, 31.0, 34.4, 35.6, 47.0, 48.0, 48.4, 51.5, and 53.0° attributed to the crystallographic planes (010), (020), (121), (022), (031), (222), (132), (230), (140), and (004), which are consistent with the reference code ICCD 98-010-5020 for a hexagonal structure with space group P 63/m.
Reviewer 2 Report
The modifications are good. It can be accepted.
Author Response
We appreciate the positive comment.
Reviewer 3 Report
Dear authors! Thank you for improving the article. However, there are still some points that should be clarified/improved.
XRD: cards assignments for all the components should be presented (for TCP, HA and chitosan), with reference database indication.
Looking at the diffractograms of the composites, I don't see peaks that could be attributed to hydroxyapatite; they should be noted.
Line 228: "These" sould be "these"
The article is templated in the Molecules journal template. It is correct?
Author Response
We appreciate the comments from the reviewer. Here are the comments, point by point, for better comprehension:
Reviewer 3
XRD: cards assignments for all the components should be presented (for TCP, HA and chitosan), with reference database indication.
R// Thank you very much for the observation. The cards assignments for each of the components have already been added in the text and Figure 2 as follows:
Figure 2 shows the XRD pattern of the blocks of CS/HA/β-TCP. In the diffractogram, the characteristic diffraction peaks of the β-TCP structure were observed, which present angles 2θ at 13.6, 17.0, 21.8, 25.7, 26.5, 27.8, 29.6, 32.4, 35.6, 39.8, 41.1, and 41.7° related to reflections of the crystallographic planes (004), (110), (113), (106), (200), (202), (008), (205), (215), (217), (224), and (303), the latter being the most representative to confirm a tetragonal phase with a 76-P41 space group, according the international files JCPDF 00-020-0024 [1]. Additionally, it was observed that some HA angles overlap with those belonging to the β-TCP. In this sense, it was possible to follow the angles 2θ at 10.8, 21.8, 31.0, 34.4, 35.6, 47.0, 48.0, 48.4, 51.5, and 53.0° attributed to the crystallographic planes (010), (020), (121), (022), (031), (222), (132), (230), (140), and (004), which are consistent with the reference code ICCD 98-010-5020 for a hexagonal structure with space group P 63/m.
Looking at the diffractograms of the composites, I don't see peaks that could be attributed to hydroxyapatite; they should be noted.
R// Thank you very much for the suggestion. The assignment of HA peaks is tedious because the 2θ angles coincide with the β-TCP, so they often overlap. However, the following figure shows the assignments considering the crystallographic charts.
Figure 2. Diffractogram of the composites CS/HA/β-TCP. F1, 87.7%β-TCP/3.5%CS/8.8%Commercial HA/0.8mLCaCl2; F2, 87.7%β-TCP/3.5%CS/8.8%Commercial HA/1mLCaCl2; F3, 87.7% β-TCP/3.5%CS/8.8%natural HA /0.8mLCaCl2.
Line 228: "These" should be "these"
R// Thank you very much for the comment. The change has been made. Line 243
Reference
- Ortiz, C.H.; Aperador, W.; Caicedo, J.C. Physical properties evolution of β -tricalcium phosphate / hydroxyapatite heterostructures in relation to the bilayer number. Thin Solid Films 2022, 752, 139256, doi:10.1016/j.tsf.2022.139256.
- Kasaai, M.R. Calculation of Mark–Houwink–Sakurada (MHS) equation viscometric constants for chitosan in any solvent–temperature system using experimental reported viscometric constants data. 2007, 68, 477–488, doi:10.1016/j.carbpol.2006.11.006.
Round 3
Reviewer 1 Report
The authors have successfully responded to all the raised questions.